# Unsupervised detection of fragment length signatures of circulating tumor DNA using non-negative matrix factorization

Gabriel Renaud[1,2,3], Maibritt Nørgaard[2,3], Johan Lindberg[4], Henrik Grönberg[4], Bram De Laere[4,5,6], Jørgen Bjerggaard Jensen[7], Michael Borre[8], Claus Lindbjerg Andersen[2,3], Karina Dalsgaard Sørensen[2,3], Lasse Maretty[2,3,9]*, Søren Besenbacher[2,3,9]*

[1]Department of Health Technology, Section of Bioinformatics, Technical University of Denmark, Kongens Lyngby, Denmark; [2]Department of Molecular Medicine, Aarhus University, Aarhus, Denmark; [3]Department of Clinical Medicine, Aarhus University, Aarhus, Denmark; [4]Department of Medical Epidemiology and Biostatistics, Karolinska Institute, Stockholm, Sweden; [5]Cancer Research Institute Gent (CRIG), Ghent University, Ghent, Belgium; [6]Department of Human Structure and Repair, Ghent University, Ghent, Belgium; [7]Department of Urology, Regional Hospital of West Jutland, Holstebro, Denmark; [8]Department of Urology, Aarhus University Hospital, Aarhus, Denmark; [9]Bioinformatics Research Centre, Aarhus University, Aarhus, Denmark

*For correspondence:
lasse.maretty@clin.au.dk (LM);
besenbacher@clin.au.dk (SB)

**Competing interest:** The authors declare that no competing interests exist.

**Abstract** Sequencing of cell-free DNA (cfDNA) is currently being used to detect cancer by searching both for mutational and non-mutational alterations. Recent work has shown that the length distribution of cfDNA fragments from a cancer patient can inform tumor load and type. Here, we propose non-negative matrix factorization (NMF) of fragment length distributions as a novel and completely unsupervised method for studying fragment length patterns in cfDNA. Using shallow whole-genome sequencing (sWGS) of cfDNA from a cohort of patients with metastatic castration-resistant prostate cancer (mCRPC), we demonstrate how NMF accurately infers the true tumor fragment length distribution as an NMF component - and that the sample weights of this component correlate with ctDNA levels ($r=0.75$). We further demonstrate how using several NMF components enables accurate cancer detection on data from various early stage cancers (AUC = 0.96). Finally, we show that NMF, when applied across genomic regions, can be used to discover fragment length signatures associated with open chromatin.

## Editor's evaluation

The authors introduce non-negative matrix factorization to analyze shallow WGS sequencing data to detect cell-free DNA fragments diagnostic of cancer. This is an area of active research and the authors add a potentially very useful unsupervised approach to analyze such data.

## Introduction

Circulating cell-free DNA (cfDNA) is rapidly emerging as an important biomarker – most notably in cancer and pregnancy. In the cancer setting, the detection of tumor cell derived DNA fragments containing somatically acquired mutations can reveal the presence of cancer. Most approaches rely on the detection of mutations using deep, targeted sequencing of a few genomic regions known to harbor driver mutations for the cancer type of interest (*Phallen et al., 2017*). While deleterious or activating mutations in known driver genes are highly specific for cancer, the sensitivity of this approach is constrained as the cancer may not contain the mutation – or the mutations may not be detectable in the blood sample due to low concentration of circulating tumor DNA (ctDNA) (*Bettegowda et al., 2014*).

It is, however, possible to get more information out of cfDNA data than just genetic variants. A key difference between cfDNA data and ordinary sequence data is that cfDNA is fragmented in vivo by a combination of enzymatic and non-enzymatic processes. Most importantly, during apoptosis, DNA is enzymatically cut between nucleosomes, and hence the lengths and positions of cfDNA fragments reflect the epigenetic state of the cell-types of origin (*Ulz et al., 2016*; *Snyder et al., 2016*). Furthermore, other enzymatic and non-enzymatic fragmentation processes (e.g. oxidative stress) may further contribute to cancer associated fragmentation patterns (*Heitzer et al., 2020*). In contrast to mutations, these signals are expected to occur across the entire genome and suggest that a focus on sequencing width instead of depth can improve sensitivity.

The fragment length distribution has been a major focus of studies searching for non-mutational signals in cfDNA. Early studies used quantitative PCR with competitive primer sets targeting amplicons of different lengths to study the cfDNA fragment length distribution in cancer using either human-mouse xenografts or blood cfDNA representing different clinical states (*Wang et al., 2003*; *Umetani et al., 2006*; *Chan et al., 2008*; *Mouliere et al., 2011*). Yet, while cancer was clearly associated with changes in cfDNA fragmentation across all studies, conflicting results were obtained with respect to the direction of change (shortening or lengthening).

A more detailed picture of how cancer manifests in the fragment length distribution of cfDNA was later obtained using short-read sequencing based approaches. First, Jiang et al. used sequencing to determine that hepatocellular carcinoma is associated with a left shift in the cfDNA fragment length distribution (*Jiang et al., 2015*). Later on, Mouliere et al. used exome and shallow whole-genome sequencing (sWGS) to investigate cancer-specific cfDNA fragmentation patterns using human-mouse xenografts or cancer mutations to separate cancer fragments from background cfDNA (*Mouliere et al., 2018*). Again, they observed a number of cancer-specific distortions including left-shifting of both the mono- and di-nucleosome peaks, and a more prominent di-nucleosome peak, and used these to accurately discriminate cancer patients from healthy controls. Finally, Sanchez et al. also applied WGS to study differences in the fragment length distribution between cases and controls, and demonstrated significant differences in fragment lengths between single- and double-stranded cfDNA populations (*Sanchez et al., 2021*).

In their *DELFI* approach, Cristiano et al. added a genomic dimension to cfDNA fragment length analyses by computing the ratio of short (100–150 bp) to long (151–220 bp) fragments in 5 MB windows along the genome and showed how these capture nucleosomal distances, which in turn reflect chromatin state (*Cristiano et al., 2019*). Using these fragment length profiles as inputs to a machine learning classifier enabled accurate discrimination of earlier stage cancers from controls.

In this manuscript, we introduce a new computational method based on non-negative matrix factorization (NMF), an unsupervised learning method, for simultaneously determining the contributions of different cfDNA sources (e.g. background and tumor) to a sample along with the fragment length signatures of each source (*Lee and Seung, 1999*). The method is completely unsupervised and uses only fragment length histograms as input and hence provide estimates that do not require xenografting approaches, knowledge about genomic alterations (e.g. SNV, CNV) or sample information like disease state. Software for calculating fragment lengths and applying NMF is available under the MIT license (*Renaud, 2022a*; copy archived at swh:1:rev:cf9ed4240b74c866f62b3da2cdb4f0bbceb7f551).

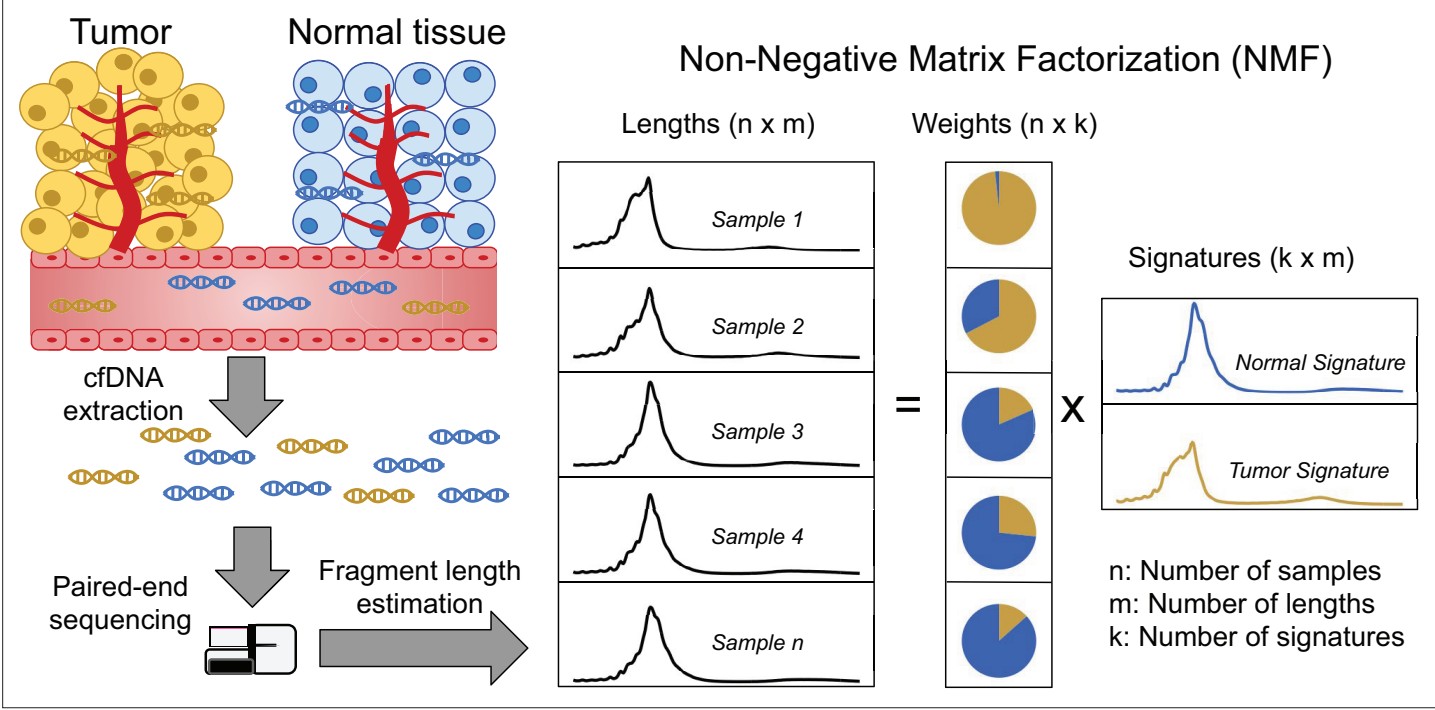

**Figure 1.** Discovering fragment length signatures using non-negative matrix factorization. The cell-free DNA (cfDNA) pool contains a mixture of fragments from different sources such as tumor cells and background (mainly cells of hematopoietic origin). After performing paired-end sequencing of cfDNA, we estimate fragment length histograms for each sample by aligning reads to the reference genome. We next generate a matrix with fragment length frequencies such that rows and columns represent samples and fragment lengths, respectively. After normalizing the rows of this matrix, we then factorize it into two non-negative matrices: (1) The *signature* matrix is aligned with columns and expresses the preference of each cfDNA source for different fragment lengths and (2) the *weight* matrix, which is aligned with rows, and contains the estimated contributions of each source to each sample.

## Results

### Discovering tumor fragment length signatures using non-negative matrix factorization

Our approach begins by computing cfDNA fragment length histograms in a series of samples based on paired-end read alignments and then uses these to construct a matrix with cfDNA fragment counts such that each row corresponds to a sample and each column to a specific fragment length (see *Figure 1* for a schematic representation of the workflow). We then normalize the rows of this matrix such that they sum to one before performing NMF, where the input matrix is approximated as the product of two non-negative matrices – both smaller than the input. One of these matrices, the *signature* matrix, has as many columns as the original matrix and represents the preference of observing each fragment length for each cfDNA source. The other matrix, the *weight* matrix, has as many rows as the input matrix and represents the contributions of each cfDNA source to each sample. The number of cfDNA sources is a hyperparameter that needs to be set in advance.

We first tested our method on sWGS of cfDNA (coverage mean: 0.60 X, range: 0.36 X–0.93X, median #read-pairs: 19.9 M) in 142 plasma samples from 94 patients with metastatic castration resistant prostate cancer (mCRPC). Samples were collected either before the initiation of first line treatment (*n*=93) or at disease progression after first line treatment (*n*=34) or later treatments (*n*=15) with some individuals (*n*=36) sampled at multiple timepoints. The observed fragment length distributions of high and low ctDNA samples differed (*Figure 2a*) and resembled those of previous reports (e.g. *Jiang et al., 2015*; *Mouliere et al., 2018*; *Sanchez et al., 2021*). Assuming two cfDNA sources (tumor/non-tumor), we then estimated fragment length signatures and weights using NMF on the normalized table of fragment length frequencies. One of the signatures (signature#2) recapitulates key features of the previously described tumor signatures including left skew, increased 10 bp periodicity left

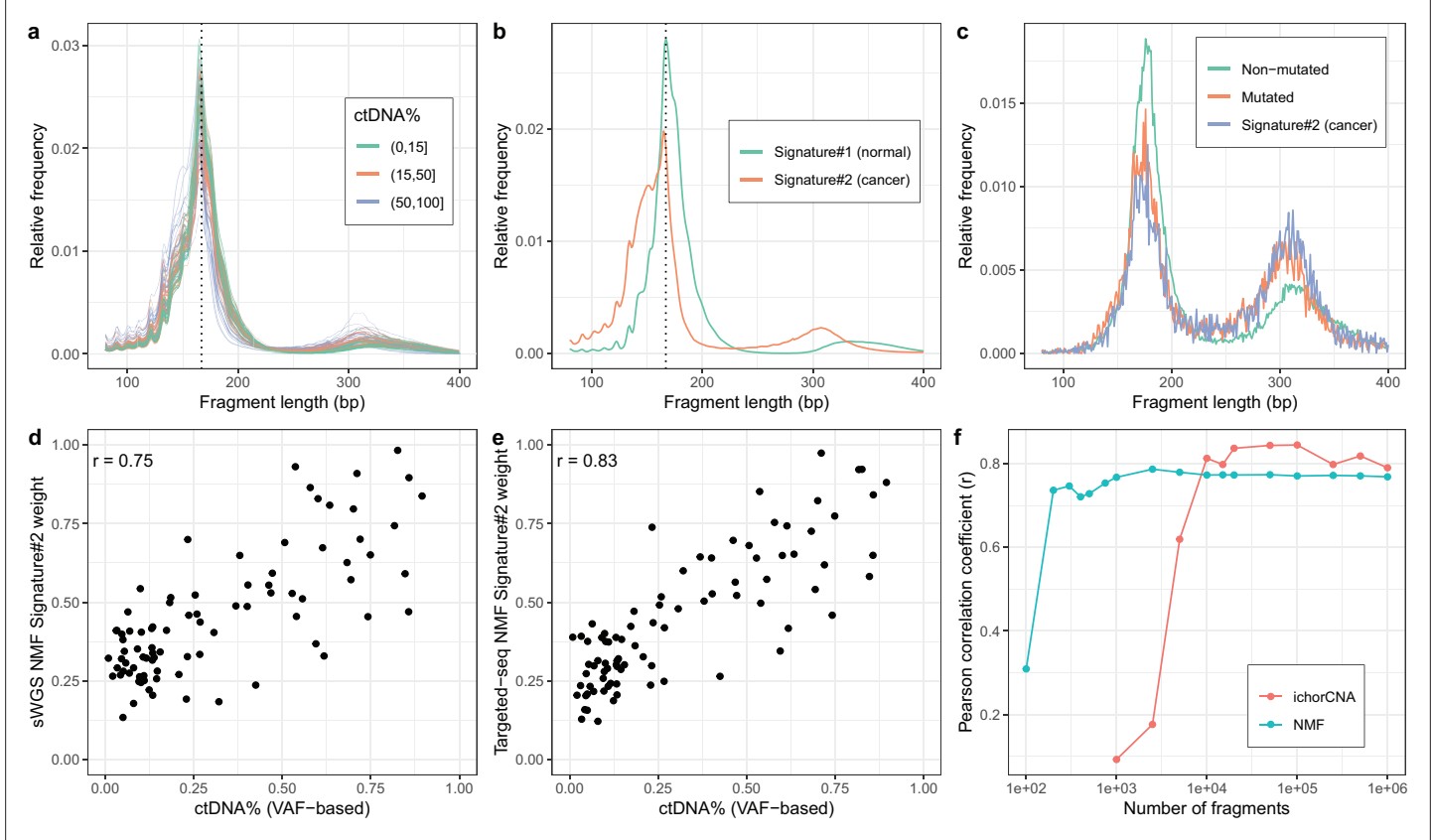

**Figure 2.** Non-negativematrix factorization (NMF) on shallow whole-genome sequencing (sWGS) and deep targeted sequencing of cell-freeDNA (cfDNA) from prostate cancer patients. (**a**) sWGS fragment length histograms for 86 prostate cancer patients; colors reflect ctDNA fractions estimated from driver variant allele fractions obtained from targeted sequencing performed on the same samples. (**b**) Fragment length signatures inferred using NMF with two components on the sWGS dataset. (**c**) Lengths of fragments containing a driver mutation (red dots), lengths of fragments overlapping the mutated position but not containing the mutation (green line) and tumor fragment length signature estimated by NMF (blue line). (**d**) ctDNA fractions estimated using driver allele frequencies from targeted data versus weights of the second NMF component estimated on sWGS data (signature#2 in panel b). (**e**) ctDNA fractions estimated using driver allele frequencies from targeted data versus weights of the second NMF component estimated on the same targeted data. (**f**) Correlation of tumor signature weights estimated using NMF and ichorCNA with ctDNA fractions for different levels of downsampling of the sWGS data.

The online version of this article includes the following figure supplement(s) for figure 2:

**Figure supplement 1.** Distribution of sample ctDNA% in the metastatic castration-resistant prostate cancer (mCRPC) cohort.

**Figure supplement 2.** Optimal number of non-negative matrix factorization (NMF) components on sWGS data from the metastatic castration-resistant prostate cancer (mCRPC) cohort.

**Figure supplement 3.** Ratio of short (100–150 bp) to long (151–220 bp) fragments vs ctDNA% (VAF-based) on shallow whole-genome sequencing (sWGS) data from the metastatic castration-resistant prostate cancer (mCRPC) cohort.

**Figure supplement 4.** ichorCNA ctDNA% estimates vs ctDNA% (VAF-based) on shallow whole-genome sequencing (sWGS) data from the metastatic castration-resistant prostate cancer (mCRPC) cohort.

**Figure supplement 5.** Stability of NMF fragment length signatures.

**Figure supplement 6.** Correlation of non-negative matrix factorization (NMF) tumor signature weights with ctDNA% on hold-out data.

**Figure supplement 7.** Overview of the analyses on metastatic castration-resistant prostate cancer (mCRPC) data presented in *Figure 2*.

of the main mode and an enlarged second peak suggesting that this source represents the tumor (*Figure 2b*).

To confirm the ability of NMF to separate tumor and background (i.e. non-tumor) sources, we performed targeted, deep sequencing (coverage mean: 647 X, range:152X–1198X, median #read-pairs cfDNA / PBMC: 35.0 M / 6.70 M) for a subset of 86 mCRPC patients using a panel of genes related to prostate cancer (5137 regions, ~1.2 MB) (*Mayrhofer et al., 2018*; *Crippa et al., 2020*).

We then called somatic variants in this data and estimated NMF using only fragments that overlap a mutated position. The putative cancer signature could then be compared to the fragment length distribution of fragments with and without mutations (*Figure 2c*). The fragment length distribution of fragments containing mutations closely matched the suspected tumor signature estimated using NMF and hence confirms that our method is able to separate tumor and background cfDNA sources solely based on fragment length information.

We next sought to investigate whether the estimated tumor signature weights are related to the blood ctDNA fraction. We, therefore, compared the sample weights of the tumor fragment length signature against sample ctDNA fractions estimated based on driver variant allele fractions (VAFs) obtained from the targeted sequencing data (min, median, max: 0.009, 0.197, 0.895) as well as estimates obtained using ichorCNA (min, median, max: 0.030, 0.071, 0.728), which uses CNVs (*Figure 2—figure supplement 1*). The estimated weights correlate strongly with ctDNA fraction for both sWGS ($r=0.75$, *Figure 2d*) and targeted sequencing data ($r=0.83$, *Figure 2e*) and hence both confirm the tumor origin of the signatures and demonstrate that our method can be used to estimate ctDNA fraction in the absence of any variant (i.e. SNVs/indels/CNVs) or clinical information.

We also explored using more cfDNA sources in the NMF. More specifically, we ran NMF for up to four components and empirically tested adding the weights of different combinations of NMF components (i.e. assuming different tumor sources) by comparing them with the driver VAF-based estimates (*Figure 2—figure supplement 2*). Only marginal improvements were observed using the more complex models suggesting the simpler alternative with two cfDNA sources is preferable as it is more interpretable and leaves little room for overfitting. In resumé, our method is able to determine fragment length signatures and ctDNA fractions on both sWGS and panel sequencing data in a completely unsupervised manner.

The performance of our method was superior to using the ratio of short-to-long fragments (100–150 bp vs. 151–220 bp) proposed by Cristiano et al. and analogous to the PCR-based methods as judged against driver VAF-based ctDNA fraction estimates ($r=0.68$, *Figure 2—figure supplement 3*). ichorCNA, which is based on CNV signals, attained better performance than NMF ($r=0.79$, *Figure 2—figure supplement 4*). We speculated that our method might work better when the data is sparse as it leverages information across the entire genome rather than only regions affected by CNVs. We thus ran both NMF and ichorCNA on the sWGS data for different levels of subsampling (*Figure 2f*). We observed that, while ichorCNA correlated better with the driver VAF-based ctDNA fractions for higher depths, our method seemed markedly better at low depths. Intriguingly, our method attained correlations greater than 0.68 with the VAF-based estimates using as little as 1000 fragments per sample. We further wished to assess how many samples are required to obtain reliable estimates using NMF. We therefore repeatedly constructed new datasets by randomly selecting a subset of samples and then training NMF models on the reduced datasets. We then computed cosine similarities between signatures estimated on the reduced datasets and those estimated on the entire dataset. Using this approach, we found that the correct fragment length signatures could be estimated with as little as 20 samples (*Figure 2—figure supplement 5*). Finally, to determine our model's ability to generalize on unseen data, we performed repeated experiments with training the model on half of the samples and predicting only the signature weights on the remaining samples. Here, we observed similar ctDNA% correlations between the samples used for training relative to those that were held out (*Figure 2—figure supplement 6*).

## Detecting cancer using two fragment length signatures

To see if fragment length signatures can detect the presence of cancer fragments from different cancer types, we reanalyzed the data from the DELFI study (*Cristiano et al., 2019*). We obtained the raw sWGS data (mean coverage: 2.84X range: 0.71X–13.4X) from 498 samples from this study including 260 healthy controls and 238 cancers distributed across seven different cancer types. The fragment length distributions in these samples (see *Figure 3a*; *Figure 3—figure supplement 1*) were similar to the prostate cancer data, and as expected, we saw a general tendency for shorter fragment lengths in cases compared to controls. There were, however, also visible differences between the prostate data and the DELFI data set. We observed, for instance, fewer di-nucleosome fragments in both DELFI cases and controls compared to the prostate cancer data. Like the prostate analysis, we trained NMF on the matrix containing fragment length histograms for each sample, again assuming two cfDNA

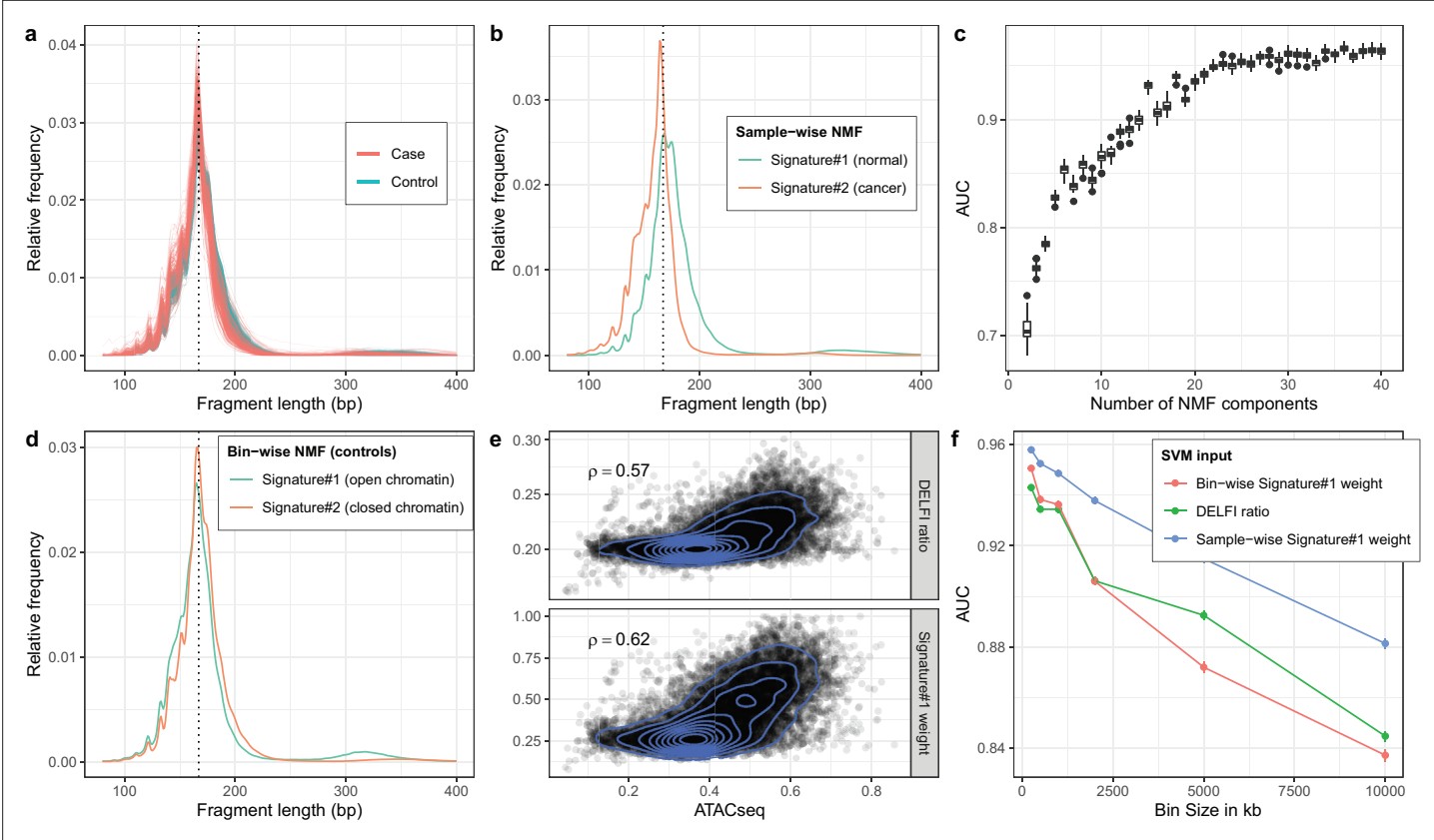

**Figure 3.** Non-negative matrix factorization (NMF) on cell-free DNA (cfDNA) shallow whole-genome sequencing (sWGS) from the DELFI study. (**a**) sWGS fragment length histograms for the 533 DELFI samples; colors indicate case-control status of the sample. (**b**) Fragment length signatures inferred using NMF with two components on the sWGS dataset. (**c**) AUCs obtained when discriminating cases versus controls using a linear Support Vector Machine (SVM) on the sample component weights across different numbers of components in the NMF model. Boxplots are based on repeating the Cross Validation 50 times. (**d**) Chromatin state fragment length signatures estimated using fragment length histograms from 250 kb bins along the genome aggregated across all control samples. (**e**) Ratio of short (100–150 bp) to long (151–220 bp) fragments ('DELFI ratio') or weight of the first NMF component (signature#1 in panel d) versus ENCODE ATACseq from a Lymphoblastoid cell-line for 250 kb genomic bins. (**f**) AUCs obtained when discriminating cases versus controls using a linear SVM on 'DELFI ratio' or weight of the first NMF components from panel **d** (red) or panel **b** (blue) inferred in bins along the genome for different bin sizes.

The online version of this article includes the following figure supplement(s) for figure 3:

**Figure supplement 1.** Shallow whole-genome sequencing (sWGS) fragment length histograms for the 533 DELFI samples stratified by stage; 'H' indicates healthy controls.

**Figure supplement 2.** Using non-negative matrix factorization (NMF) with two fragment length signatures for classification on DELFI data stratified by cancer type.

**Figure supplement 3.** Cancer type specific non-negative matrix factorization (NMF) models.

**Figure supplement 4.** Using a Support Vector Machine (SVM) trained on 30 non-negative matrix factorization (NMF) components to classify sequential samples.

**Figure supplement 5.** Overview of the analyses on the DELFI dataset presented in *Figure 3*.

sources (see *Figure 3b*). We assumed that the signature with the lower mean fragment length was the cancer-related signature. This signature did indeed tend to have a higher weight in the cancer samples (*Figure 3—figure supplement 2a*), and it could differentiate between cancer and control samples with an AUC of 0.75 (*Figure 3—figure supplement 2b*). We also investigated if the AUC differed when restricting the input data to a specific cancer type. The AUC was highest for colorectal cancer patients (0.93) and lowest for gastric cancer (0.56). Full results are shown in *Figure 3—figure supplement 3a*. Furthermore, we sought to determine whether the inferred signatures differ from one cancer type to the next. The different inferred signatures were plotted (*Figure 3—figure supplement*

*3b*). Apart from gastric cancer, which happens to be the one with the lowest AUC, most fragment length signatures do not seem to change depending on the cancer type.

## Detecting cancer using more than two fragment length signatures

We then tested whether using more signatures would improve this classification. We used a linear Support Vector Machine (SVM) to see if we could separate cancer and control samples based on their signature weights for a given number of signatures. The results showed that adding more signatures significantly improved the AUC (see *Figure 3c*). The classification continued to get better until we reached ~30 signatures, and for 30+signatures, we got an AUC above 0.95. This AUC is comparable to the AUC of 0.94 the DELFI method achieves by using Gradient Boosting Machine on ~500 features per sample. When testing a SVM with 30 signatures on the sequential samples from the DELFI article a majority of the samples had a positive correlation between the predicted case probability and the tumor fraction estimated based on VAF (*Figure 3—figure supplement 4*).

## Estimating chromatin state signatures using NMF

The DELFI method also uses fragment length information to detect cancer samples, but rather than looking at the total length distribution, it looks at fragment lengths in bins along the genome (*Cristiano et al., 2019*). The DELFI creators have shown that the length distribution of cfDNA fragments in a genomic region carries information about its chromatin state (open or closed) and that this information can be used to distinguish cancer samples from healthy controls. Specifically, the DELFI method uses the ratio of short (100–150 bp) to long (151–220 bp) fragments in 5 MB windows along the genome as input to a machine learning classifier. We wished to investigate whether we could use NMF fragment length signatures to better capture this chromatin state and hence yield better classification. First, we partitioned the genome into non-overlapping bins of 250 kb and computed the fragment length histograms for each bin in each sample. We then summed histograms for each bin across all healthy controls to yield a matrix with rows corresponding to genomic bins instead of samples as before, and ran NMF using two signatures. This resulted in the two *bin-wise* NMF signatures shown in *Figure 3d*. To compare, we also calculated the short-to-long ratio used by the DELFI method in each bin. Looking at chromatin accessibility in a lymphoblastoid cell-line measured by ATACseq for the 250 kb bins, we observed a slightly better Spearman correlation for the weight of the first signature than the DELFI ratio (*Figure 3e*).

## Using regional signature weights for classification

We then wanted to test whether using the NMF weight in each bin as input features for a classification method would be superior to using the DELFI ratios. We estimated the first bin-wise signature's weight in each bin in each sample and used these weights as input to an SVM and assessed classification performance across different bin sizes. The results in *Figure 3f* show that for small bin sizes, we get a slightly better result by using the bin-wise NMF weight for each bin instead of the short-to-long DELFI ratio as SVM input. But for larger bin sizes, the DELFI ratio is better than using the bin-wise NMF signature. Finally, we also tried inferring the weight of the sample-wise NMF signature from *Figure 3b* for each bin and using that as input to the SVM (blue line in *Figure 3f*). This turned out to give superior results compared to using the bin-wise NMF signature.

## Discussion

In this article, we propose non-negative matrix factorization (NMF) of fragment length distributions as a new tool in the cfDNA-seq analysis toolbox. A key objective in cfDNA-seq analyses in the cancer setting is to determine the amount of circulating tumor DNA – most importantly whether any tumor DNA is present at all. This carries immense importance both for early detection of cancer in the screening setting – and for detection of relapse after treatment, due to residual disease after surgery or resistance to chemotherapy.

Most analyses have focused on using mutational signals for determining ctDNA load through estimation of variant allele fractions for SNVs in deep, targeted cfDNA-seq data (*Phallen et al., 2017*) – or CNVs by sWGS (*Adalsteinsson et al., 2017*). Yet non-mutational signals such as those shown by several studies to be manifested in the fragment length distribution are gaining traction as they

may improve sensitivity by enabling aggregation of the cancer signal across the entire genome rather than just positions affected by mutations. So far, fragment length signals have been approached either using a simple summary statistic like the ratio of short to long fragments e.g. DELFI (*Cristiano et al., 2019*) or by manually curating features reflecting the distribution and then use these as input for a supervised machine learning algorithm (*Mouliere et al., 2018*). However, manual featurization may not make the best use of all relevant information contained in the distribution – and supervised learning with many features carries the risk of overfitting.

We propose NMF as a general analytical framework for working with cfDNA-seq fragment length distributions. NMF enables us to simultaneously estimate fragment length *signatures* and their *weights* in each sample. Using sWGS cfDNA-seq data from a cohort of patients with metastatic prostate cancer, we show that NMF estimated with two components discovers a signature, which accurately matches the true tumor fragment length distribution and exhibits many of the characteristics previously associated with ctDNA. The weights of this signature correlated strongly with ctDNA levels – nearly as good as ichorCNA - without using any information about variants or ctDNA levels. Importantly, similar results were obtained when using deep, targeted cfDNA-seq. Furthermore, subsampling experiments revealed that NMF was markedly more robust when less data is available than ichorCNA. This may indicate that NMF is more robust than ichorCNA at lower tumor fractions. However, the lack of low ctDNA samples is a limitation of the current study and further experiments either based on samples with lower tumor burdens, spike-ins or in silico dilution are needed to confirm this assertion as it could not be directly tested with the data available in this study. A lack of healthy controls for the mCRPC cohort data meant that we were not able to perform spike-in or in silico dilution experiments in this study. Finally, as the fragment length signal is likely orthogonal to any mutational signal, it may be possible to combine these lines of information to obtain a better, joint estimate of tumor load.

The unsupervised nature of NMF implies little risk of overfitting and the ability to inspect the fragment length profiles of each signature provides full transparency of the method in contrast to many other approaches such as the supervised learning by Random Forest strategy applied by Mouliere et al. for determining ctDNA levels. Transparency is important because using non-mutational information carries the risk of using information that is not directly linked to the presence of ctDNA in blood, but instead reflects, e.g. an ongoing immune response. This in turn may impact a models' ability to generalize to unseen data and clinicians' trust in the model. Using NMF, we were able to verify that the fragment length profile of the signature correlating with ctDNA levels does indeed match the length distribution of fragments containing mutations and hence that the model is directly measuring ctDNA load. Hence, the combination of unsupervised learning and transparency suggest that NMF constitutes a robust modeling framework for cfDNA-seq length spectra.

The data from the prostate cancer cohort generally contained patients with a high tumor burden and we wished to also test the models applicability in a screening context characterized by low ctDNA load and also look at different cancer types. We obtained access to the data from the DELFI study, which contains cfDNA-seq data from a range of cancer types and primarily from patients with early stage disease (*Cristiano et al., 2019*). The signatures estimated from high ctDNA load prostate cancers and those of DELFI cases shared features (e.g. left skew of the main mode), but also differed as for instance the second mode of the distribution was less pronounced in the DELFI data. These differences may reflect differences in sample processing (e.g. DNA extraction method) and sequencing technology rather than actual biological differences between the studies suggesting that transferring models trained on one dataset to another may be difficult although this assertion was not directly tested in the present study. Hence, this constitutes an important, potential limitation of 'fragmentomics' that will require further attention in the future. That said, using an unsupervised method such as NMF to learn the relevant fragment length signatures can help alleviate such transferability problems as the model can easily be retrained on, e.g. a new batch of data. To know which of the two signatures learned corresponds to cancer fragments, one could use a previous set of signatures trained on a labeled set as the starting point for the NMF optimization.

We note that a number of technical and biological factors in the DELFI study design may have inflated our classification performance. For instance, cases were generally older than controls and hence the observed differences in the fragment length distribution between cases and controls may partially reflect underlying comorbidities associated with higher age.

On the DELFI dataset, we furthermore demonstrated that using several NMF components enabled accurate cancer detection of early stage cancers – on par with the original DELFI results. We obtained the best classification results by using 30 or more signatures, and an even larger number of signatures could likely be relevant for larger or more heterogeneous datasets. Indeed, we speculate that the difference in gain from using more signatures between the mCRPC cohort, where two signatures worked well, to the DELFI study reflects a larger degree of heterogeneity in the multi-cancer DELFI study. Even when more components are required, using a supervised model with tens of parameters rather than hundreds as in the genomic window model makes overfitting less likely.

Finally, we investigated whether the NMF approach could improve upon the genomic bin-based approach proposed by Cristiano et al. We first showed how NMF can discover fragment length signatures of different chromatin states when trained across genomic bins, where the fragment length histogram in each bin has been aggregated across multiple samples (*bin-wise* training). The learned *chromatin state* signatures turned out to correlate better with open chromatin as measured using ATACseq than the DELFI ratio, but did not yield a clear improvement in classification performance over the DELFI method. We speculated that the lack of classification improvement could be due changes between cases and controls not related to chromatin status. We therefore investigated whether the bin-based approach could be improved by instead inferring the signature weights of the *sample-wise* trained NMF model in each genomic bin. To our surprise, this model outperformed both the bin-wise NMF and the DELFI ratio across all bins sizes, which may indicate that the DELFI classification signal is not purely a chromatin state signal but in part caused by CNVs in the tumors or other cancer-specific distorsions.

## Conclusions

In resumé, we here demonstrate the use of NMF as a general and robust statistical approach for analyzing fragment length distributions from cfDNA-seq.

## Materials and methods

### Sample processing and DNA extraction

A total of 142 EDTA-blood samples were collected from 94 patients with metastatic castration-resistant prostate cancer at Aarhus University Hospital and Regional Hospital of West Jutland between April 2016 and August 2019. Samples were collected either before the initiation of first line treatment ($n$=93) or at disease progression after first line treatment ($n$=34) or later treatments (2nd– 4th lines, $n$=15) with 36 patients having multiple samples taken (of these: median 2 samples/patient, range: 2–4 samples/patient).

Blood samples were collected in BD Vacutainer $K_2$ EDTA tubes (Beckton Dickinson) and processed within 2 hr (stored at 4°C until processing). To separate plasma from cellular components, EDTA blood samples were centrifuged at 2000–3000 $g$ for 10 min (20°C) and plasma stored in cryo tubes (TPP) at –80°C until cfDNA extraction. Plasma samples (2.0–4.5 ml) were thawed at room temperature and centrifuged at 3000 $g$ for 10 min (20°C). cfDNA was extracted on a QIAsymphony robot (Qiagen) using the QIAamp Circulating Nucleic Acids kit (Qiagen) as described by the manufacturer. Extracted cfDNA was stored in LoBind tubes (Eppendorf AG) at –80°C until further analysis (<1 month). cfDNA concentration was determined by droplet digital PCR (ddPCR) using a QX200 AutoDG Droplet Digital PCR System (Bio-Rad) according to the manufacturer's instructions as previously described (*Reinert et al., 2016*). Germline DNA from buffy coats [peripheral blood mononuclear cells (PBMC)] was extracted on a QIAsymphony robot (Qiagen) using the QiaSymphony DSP DNA Mini Kit (Qiagen) following manufacturer's instructions. DNA concentrations were determined using Qubit fluorometric quantification (Qubit dsDNA Broad range, Thermo Fisher Scientific). Prior to library preparation, the Covaris E220 Evolution ultrasonicator (Covaris) was used to shear germline DNA to shorter fragments (~250–350 bp).

### Library preparation and shallow whole-genome sequencing

Libraries for next generation sequencing were prepared using the Kapa Hyper Library Preparation Kit (KAPA Biosystems) with 7.0–50.0 ng cfDNA (median 30.5 ng) or 50 ng for germline DNA as input. xGen CS-adapters – Tech Access (IDT-DNA) with Unique Molecular Identifiers (UMIs) on both strands

and primers containing unique indexes on both strands were used to generate indexed libraries. Plasma libraries were pooled equimolarly and paired-end sequenced (2 × 151 bp) on an Illumina Novaseq instrument (S-prime flowcell), generating 12.66–34.30 million read pairs/ sample (median: 19.99) corresponding to coverages from 0.36 to 0.93X (median: 0.60X). Fastq files were demulti-plexed using bcl2fastq (v2.20.0.422) and quality checked using fastQC and fastqScreen (Available from: http://www.bioinformatics.babraham.ac.uk/projects).

## Estimation of fragment lengths

We used leeHom with the '--ancientdna' option on the raw fastq files to strip adapters, and, where possible, to reconstruct DNA fragments by merging overlapping paired-end reads. UMIs were removed by trimming the first five nts of reach read (*Renaud et al., 2014*). Reads were then mapped to hg19 using BWA-MEM v.0.7.17 with seed length ('-k') set to 19 (*Li, 2013*). PCR and optical dupli-cates were removed using samtools rmdup using option '-s' for single-end reads. Paired-end reads with mapping quality below 30, mapping within ENCODE excluded regions (ENCFF001TDO) or which contained soft or hard clips in either of the two reads were filtered away before computing fragment lengths. Fragment lengths were then directly obtained as the length of the reconstructed fragment when reconstruction was possible and otherwise obtained from the insert size calculated by the aligner based on distances on the reference sequence. Finally, fragment lengths less than 30 or greater than 700 were discarded and a matrix with fragment length counts constructed such that rows and columns corresponded to samples and fragment lengths, respectively.

## Non-negative matrix factorization

The rows of the matrix with fragment length counts were first scaled such that they sum to one. NMF of the normalized matrix was then performed using scikit-learn (sklearn.decomposition.NMF) with random initialization, multiplicative updates and the Kullback–Leibler loss function. For the bin-wise NMF experiments, the NMF was repeated across 20 different random initializations and the fit achieving the lowest loss selected. The estimated fragment length signature and weight matrices were scaled to sum to one for each signature and sample, respectively. Software for calculating frag-ment lengths and applying NMF is available under the MIT license (*Renaud, 2022a*).

## Subsampling experiments

To determine the sequence coverage required to estimate tumor fraction using NMF (*Figure 2f*) we subsampled input bam files using a custom C++program (https://github.com/grenaud/libbam/blob/master/subsamplebamFixedNumberSingleFirstMate.cpp; *Renaud, 2022b*; copy archived at swh:1:rev:c210c1ce1d4f0e1fe07fcbb4a438f9a7a2e3cf9b). Briefly, this program subsamples a specific number of fragments while taking into account paired-end information. The subsampled bam files were then used as input for NMF and ichorCNA and for each subsampling level, we calculated the average Pearson correlation between the NMF/ichorCNA tumor fraction estimates and the VAF based tumor fraction estimate.

To determine the number of samples required to reliably estimate fragment length signatures (*Figure 2—figure supplement 5*), we repeatedly sampled a given number of samples across a range of dataset sizes and trained NMF models on each of the generated datasets. For each trained model, we then calculated the maximum cosine similarity between signatures estimated on the smaller dataset with those estimated using the full dataset. For each number of samples, we conducted 100 replicates as to have an average behavior.

To determine accuracy on hold-out samples (*Figure 2—figure supplement 6*), we repeatedly divided the data into two partitions, a training and a test set of equal size (43 samples). For each pair, we then trained an NMF model on the training set and predicted the weights of each signature on the test set using the signatures learned from the training set. We then calculated the Pearson correlation between the estimated weights and the cfDNA% using both the weights from the training and test sets.

## Analyzing fragment lengths in bins along the genome

We first divided the genome into bins of 250 kb and estimated the fragment length distributions for each sample in each bin. We then calculated the mean number fragments in each bin across the

controls. To avoid bins with possible mapping problems, we excluded bins where the mean number of fragments in the controls were less than the median-2*IQR or greater than the median + 2*IQR. Comparison with ATACseq data was performed using ENCODE track 'ENCFF603BJO' (fold change over control, GM12878 cell line) lifted from hg38 to hg19.

## Building support-vector machine classification models

The classification results based on multiple fragment length signatures and the classification results on chromatin state signatures and DELFI ratios were calculated using linear SVMs implemented in R using tidymodels and kernlab. Before training, the values for each sample was standardized by subtracting the mean and dividing with the standard deviation. Accuracy and AUC were calculated using repeated 10-fold cross-validation (50 repeats). Scripts used for the analysis of the DELFI data are available in *Source code 3*.

## Deep targeted prostate-tailored sequencing

A total of 86 indexed libraries were subjected to deep targeted sequencing based on ichorCNA ctDNA% estimates from plasma cfDNA libraries. A previously designed gene panel that captures regions in the human genome commonly altered in PC was used for capture (*Mayrhofer et al., 2018*; *Crippa et al., 2020*). Libraries were pooled equimolarly (8-plex) for in-solution target enrichment using Twist Bioscience's Custom Target Enrichment. Captured pools were paired-end sequenced (2 × 100 bp) on an Illumina Novaseq instrument (S1 flowcell). Fastq files were generated as for the sWGS data and adaptor trimming, mapping, duplicate removal, realignment, and quality score recalibration performed as described above for the ichorCNA workflow for sWGS data. The resulting depth of coverage for the targeted regions was on average 647X with a minimum of 152X and a maximum of 1198X. The average coverage for the targeted sequencing of germline DNA from buffy coats was 306X (range: 253–728X).

## Variant calling and interpretation

Somatic variants were called using VarDict (v. 1.6) (*Lai et al., 2016*), Strelka2 Somatic (v. 2.9.10) (*Kim et al., 2018*), GATK mutect2 (v. 4.1.2.0) (*Cibulskis et al., 2013*), and VarScan2 (v. 2.4.2) (*Koboldt et al., 2012*). For each sample, a patient-matched germline sample was used as control. The impact of each variant was annotated using the Ensembl Variant Effect Predictor (ensemble-vep v. 96.0) (*McLaren et al., 2016*). Somatic variants were required to be supported by at least 10 reads, called by ≥3 callers, and annotated as either pathogenic or likely pathogenic in OncoKB or ClinVar (*Chakravarty et al., 2017*; *Landrum et al., 2014*) or introduce a premature stop or frameshift in the coding sequence. High-impact variants called by 2 callers were not discarded whereas low-frequency variants (VAFs <0.02) were discarded unless they had high impact, were called by all 4 callers, or were detected in another sample from the same individual. LOH-status was annotated for each variant using cfDNA copy number profiles and allele ratio of heterozygous SNPs. All variants were manually curated in IGV (v. 2.5.3) and we made sure that no alternative reads were seen in the germline DNA.

## Running ichorCNA

Adapter sequences were trimmed using Cutadapt (v1.16) (*Martin, 2011*) and paired-end sequences mapped to the hg19 reference genome using BWA MEM (v0.7.15-r1140) (*Li, 2013*). PCR and optical duplicates were removed from each library independently using Samblaster (v0.1.24) (*Faust and Hall, 2014*) and the final bam files realigned (GATK v3.8.1.0) (*DePristo et al., 2011*). ichorCNA was run with default parameters (*Adalsteinsson et al., 2017*).

## Estimation of ctDNA fractions from targeted seq data

CtDNA fractions were also estimated from the targeted sequencing data. First, the tumor cell purity (i.e. tumor cell fraction) was calculated from somatic mutations with moderate or high impact. In brief, for each sample the somatic mutation with the highest VAF was used (i.e. clear driver mutation). If multiple mutations had similar VAFs (±2%), the median VAF of these mutations were used instead. A total of 1–2 mutations per sample were used to estimate purity. Accounting for LOH status of the mutation(s), the tumor cell purity was estimated as follows: purity = 2*VAF (no LOH) or purity = 2/(1/VAF + 1) (LOH). Next, to obtain ctDNA fractions, the tumor cell purity was adjusted for tumor ploidy:

ctDNA fraction = tumor cell purity * tumor ploidy/(tumor cell purity * tumor ploidy + normal cell purity * normal ploidy). Here, normal purity was set to 1-tumor cell purity and normal ploidy was set to 2. Tumor ploidy estimates were obtained from PureCN (*Riester et al., 2016*).

## Acknowledgements

The authors wish to thank the staff at the departments of urology at Aarhus University Hospital and Regional Hospital West Jutland for patient recruitment and collection of clinical data. We would also like to thank lab technicians and clinical academics at the Department of Molecular Medicine (AUH, Denmark) and the Department of Medical Epidemiology and Biostatistics (Karolinska Institutet, Sweden) for excellent assistance throughout the project.

## Additional information

### Funding

| Funder | Grant reference number | Author |
| --- | --- | --- |
| Independent Research Fund Denmark | Sapere Aude Research Leader | Søren Besenbacher |
| Danish Cancer Society | | Karina Dalsgaard Sørensen |
| The Central Denmark Region Health Fund | | Karina Dalsgaard Sørensen |
| Aarhus Universitet | Graduate School of Health | Maibritt Nørgaard |
| Direktør Emil C. Hertz og Hustru Inger Hertz Fond | | Karina Dalsgaard Sørensen |
| KV Fonden | | Karina Dalsgaard Sørensen |
| Raimond og Dagmar Ringgård-Bohns Fond | | Karina Dalsgaard Sørensen |
| Beckett-Fonden | | Karina Dalsgaard Sørensen |
| Snedkermester Sophus Jacobsen og Hustru Astrid Jacobsens Fond | | Karina Dalsgaard Sørensen |

The funders had no role in study design, data collection and interpretation, or the decision to submit the work for publication.

### Author contributions

Gabriel Renaud, Conceptualization, Software, Formal analysis, Supervision, Funding acquisition, Investigation, Visualization, Methodology, Writing – original draft, Project administration, Writing – review and editing; Maibritt Nørgaard, Data curation, Software, Formal analysis, Investigation, Visualization, Writing – original draft, Writing – review and editing; Johan Lindberg, Resources, Data curation, Formal analysis, Investigation, Methodology, Writing – review and editing; Henrik Grönberg, Resources, Methodology; Bram De Laere, Jørgen Bjerggaard Jensen, Resources, Methodology, Writing – review and editing; Michael Borre, Resources; Claus Lindbjerg Andersen, Resources, Writing – review and editing; Karina Dalsgaard Sørensen, Supervision, Funding acquisition, Methodology, Project administration, Writing – review and editing; Lasse Maretty, Søren Besenbacher, Conceptualization, Formal analysis, Supervision, Funding acquisition, Investigation, Visualization, Methodology, Writing – original draft, Project administration, Writing – review and editing

### Author ORCIDs

Søren Besenbacher ⓘ http://orcid.org/0000-0003-1455-1738

### Ethics

Human subjects: The prostate study was approved by The National Committee on Health Research Ethics (#1901101) and notified to The Danish Data Protection Agency (#1-16-02-366-15). All patients provided written informed consent.

### Decision letter and Author response

Decision letter https://doi.org/10.7554/eLife.71569.sa1
Author response https://doi.org/10.7554/eLife.71569.sa2

---

# Additional files

### Supplementary files

• Supplementary file 1. Fragment length distributions in mCRPC cohort. Sheet 1 contains raw fragment length distributions from WGS data along with ctDNA% estimates. Sheet 2 contains raw fragment length distributions from targeted data.

• Transparent reporting form

• Source code 1. Source code and data to produce *Figure 2*.

• Source code 2. Source code and data to produce *Figure 3*.

• Source code 3. Scripts used for the analysis of the DELFI data. This includes a script to train NMF, a script to estimate the weight of NMF components and a script to train and evaluate a linear SVM model.

### Data availability

Danish law requires ethical approval of any specific research aim and imposes restrictions on sharing of personal data. This means that the prostate cancer data used in this article cannot be uploaded to international databases. External researchers (academic or commercial) interested in analysing the prostate dataset (including any derivatives of it) will need to contact the Data Access Committee via email to kdso@clin.au.dk. The Data Access Committee is formed of co-authors Karina Dalsgaard Sø;rensen and Michael Borre, and Ole Halfdan Larsen (Department Head Consultant, Department of Clinical Medicine, Aarhus University). Due to Danish Law, for the authors to be allowed to share the data (pseudonymised) it will require prior approval from The Danish National Committee on Health Research Ethics (or similar) for the specific new research goal. The author (based in Denmark) has to submit the application for ethical approval, with the external researcher(s) as named collaborator(s). In addition to ethical approval, a Collaboration Agreement and a Data Processing Agreement is required, both of which must be approved by the legal office of the institution of the author (data owner) and the legal office of the institution of the external researcher (data processor). Raw fragment length distributions along with ctDNA% estimates are available in Supplementary file 1.

The following previously published dataset was used:

| Author(s) | Year | Dataset title | Dataset URL | Database and Identifier |
|---|---|---|---|---|
| Karlijn L | 2019 | Genome-wide cell-free DNA fragmentation in patients with cancer | https://ega-archive.org/datasets/EGAD00001005339 | EGA, EGAD00001005339 |

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
