## [Editor Report]

The authors introduce non-negative matrix factorization to analyze shallow WGS sequencing data to detect cell-free DNA fragments diagnostic of cancer. This is an area of active research and the authors add a potentially very useful unsupervised approach to analyze such data.

---

## [Decision Letter]

**Decision letter after peer review:**

Thank you for submitting your article "Discovering fragment length signatures of circulating tumor DNA using Non-negative Matrix Factorization" for consideration by *eLife*. Your article has been reviewed by 3 peer reviewers, and the evaluation has been overseen by Y M Dennis Lo as the Senior/Reviewing Editor. The following individuals involved in review of your submission have agreed to reveal their identity: Alain Thierry (Reviewer #1).

Essential revisions:

1. More information on the source of cancer patient plasma, characteristics of the patients, especially in respect to treatments at time of blood sampling, should be included. A flow chart summarising the overall results might be useful.

2. The authors can consider citing some of the older work analyzing cfDNA fragment size distribution using PCR-based approaches.

3. The authors showed exactly the same illustrative fragment size profile in Figure 1 (normal and tumor signature) as those presented earlier by Sanchez et al. (JCI insight, 2021). Consequently, they should at least mention in a few sentences the observations made by Jiang et al. and by Sanchez et al. (JCI insight, 2021) in order to validate their assumption.

4. As one could argue that the signatures of cfDNA fragment length had been discovered through various previously published work, in light of the previous concerns, term 'discovery' is perhaps inappropriate. Hence, we would suggest replacing the word 'discovery' by another term such as 'detection' or 'assessment'.

5. In the introduction, it would be better to describe a bit more details about tumor DNA length characteristics that were reported previously, so that readers would have a more comprehensive understanding of what is already known in the prior art and why the analysis of tumor-specific length profile in this study is important.

6. The authors should compare whether an NMF component would give a better correlation with tumor DNA estimated by ichorCNA, compared with the short DNA proportion (i.e.% of plasma DNA < 150 bp) in the overall size profile.

7. It seems somewhat intriguing that the classification power between cases and controls improved as the number of components used in the NMF process. However, the authors did not see an improvement in the correlation between tumor DNA fraction and the weight of one component reflecting the tumor contribution. These observations should be discussed and clarified in a revised manuscript.

8. The AUC was calculated using repeated 10-fold cross-validation. It would be better to show the boxplot of AUC values for a given number of NMF components (Figure 3c).

9. It will be informative to estimate the minimum number of samples is required for performing an accurate NMF analysis.

10. The authors claim their model is directly measuring ctDNA load. Since an untargeted (unsupervised) model to assess the tumor fraction for monitoring purposes would be of great value, it would be interesting to see if changing levels of ctDNA can be captured by changing weights of the putative cancer Signature 1. Since their mCRPC cohort included patients with follow-up samples (at therapy start and progression) it is not clear why this has not been tested.

11. It should be mentioned that ichorCNA can only provide informative results at ctDNA fraction of 3% and higher. At higher tumor fractions the fragment signature signal might be relatively strong, explaining the good correlation, but what about samples with lower fractions? Please indicate the tumor fractions (median, min, max) assessed with ichorCNA and based on the VAFs of the identified mutations for all patients.

12. The authors claim that using NMF they were able to verify that the fragment length profile of the signature correlating with ctDNA levels does indeed match the length distribution of fragments containing mutations. However, although they included PBMCs in their mutation analysis the authors did not correct for clonal hematopoiesis (CH). Recent studies showed that up to 20% of identified mutation are related to CH and that such mutations are also found in driver genes for solid tumors. Since there might be a size difference of mutated fragment originating from the tumor or PBMCs, respectively, this statement is overrated. In case their sequencing approach was not sensitive enough to detected CH-related variants this should at least be mentioned.

13. In Figure 2c, the shift towards smaller fragment is only seen for the dinucelosomes peak but not for the mononucleosomal fragments. How do the authors explain this (if not by contamination of CH-related variants)?

14. The authors demonstrated that NMF was more robust when less data was available than the ichorCNA algorithm, but they lacked to provide convincing data that NMF as a quantitative measure is more sensitive with respect to tumor fraction. To this end, in silico dilution experiments where they computationally mixed healthy fragments and high ctDNA fraction sample(s) to a ctDNA fraction of their choice (with repetitions to achieve a more robust es-timation) should be performed.

15. If there are no CNA or tumor fraction >3%, ichorCNA can literally not be used to determine ctDNA fraction, so the feasibility of the dilution approach to benchmark ichorCNA is not clear to the reviewer (in particular for early-stage cases, in which most samples have tumor fractions >1%). Moreover, low depth can be overcome by additional sequencing whereas the tumor fraction cannot be enriched by more reads.

16. Using NMF the authors demonstrate that there are 2 major contributors to fragment length distribution and the addition of further signatures does not add anything. However, this statement is conflicting with results of the linear SVM classification case/control approach (Figure 3c). A plateau of classification accuracy is reached when using around 30 signatures, which suggests this would be a good trade-off for number of signatures to reach high accuracy (AUC).

17. Figure 3d: Interestingly, open chromatin is associated with a higher fraction of dinucleosomal fragments. How do the authors explain this? Figure 3: the reviewer is not sure what the authors want to show here with Pearson's correlation coefficient, this does not look like a linear relationship.

18. The paragraph on "epigenetic fragment length" could be better presented. I would re-phrase "epigenetic state". Both the DELFI ratio and the fragment length signature are surrogates for the chromatin state (open or closed). First the author talks about bin-wise extraction of two signatures. Why is it, that reducing bin size gives better classification results? ~50 kb bins seem to be best to predict ATACseq results. Is there literature that describes open/closed chromatin in this way? Then they correlate the weights from that with ATACseq results. And finally, they use their bin-wise signature extraction results/weights as input for cancer/control classification. Could be split in two parts for clarity: one that talks about correlation with ATACseq results and one that discusses classification results.

19. cfDNA fragment length profiles are particularly prone to pre-analytical (extraction, library prep) and analytical (clustering efficiency of various sequencings machines) factors. How do preanalyitcal/analytical factors affect the classification? The authors mention in the discussion that changes in the size profiles from their mCRPC cohort and the DELFI samples vary, which might be attributed due to such factors. Yet such confounding event will impact the robustness of such approaches and may prevent a widespread implementation of fragmentomics. This should be further discussed.

20. How can the authors exclude that the above-mentioned differences are attributed to higher/lower ctDNA fractions in the metastatic CRPC and the early stage DELFI samples? Even in Figure 2a it seems that samples with higher tumor fraction have more dinucleosomal fragments. Separate fragmentation profile plots for each stage should be shown; it would be interesting whether such profiles show any differences for stage I and stage II patients.

21. The authors could have checked in the DELFI cohort whether there is a (physiologic fragment length variation) between the seven cancers types? Would there always be the same two signatures coming up if the only consider one tumor type? Based on Supplementary Figure 4 this seems not to be case. In addition, there is a huge overlap of healthy individuals and cancer patients for signature 1? Please also present Supplementary Figure 4 with Signature 2. Can we really assume that this signature is the one related to cancer?

22. Abbreviations should be defined at the beginning for better readability.

23. Resolution of figures need to be improved. Comprehensive legends should also be included for Supplementary figures.

24. Matching colors for better readability are recommended.

25. Specify average read pairs for cfDNA samples and corresponding PBMCs.

26. Excel Supplementary file: Supplementary table 2 should be mentioned in the manuscript.

27. ctDNA (%) on the x-axis in Figures 2 is calculated from targeted mutation analysis, right? How was est.ctDNA % in the Supplementary figures 2&3 calculated?

28. What is the original h matrix in NMF in Figure 2f? There is a normal h matrix described as weight matrix, but I haven't seen the definition of what with "original" h matrix is meant.

29. Assumptions for linear correlation using Pearson are not mentioned, I don't know whether they are met at all. Looking at the figure it does not seem like they are met. Normality of the variables should at least be checked.

30. How is NMF performing on previously unseen data? This part is missing. The authors could have checked for overfitting in this way for example calculating the reconstruction error on a test dataset.

31. It is not clear whether cross validation was used only for the classification task or also to evaluate the NMF decomposition. Are those signatures representative?

32. It would be nice to see the position of the excluded bins based on +- 2* IQR. They already excluded the ENCODE blacklist regions. Other outliers may be real biological signal especially in some type of cancer where copy number alteration play an important role.

*Reviewer #1 (Recommendations for the authors):*

Renaud et al. aims at developing a new computational method based on NMF relying on the difference in the fragment size profiles of cfDNA extracted from plasma of healthy individuals and metastatic castration-resistant prostate cancer patients. The manuscript is clear and the drawings very helpful. While data are convincing, the reviewer has a few major concerns that should be necessarily addressed:

*Major concerns:*1. The authors generally poorly refer to the source of cancer patient plasma. Limitations of the work, especially method performance, is slightly described in particular about cancer stages, or other cancers. In addition, characteristics of the patients are missing, especially in respect to treatments at time of blood sampling. Lastly, a flow chart is missing, enabling to evaluate successful rates.

2. Citation of critical reports or issues are missing when introducing this work and placing it into the development of methods for pan cancer screening. Below are concerns that need to addressed:

– The authors are citing reports on cancer vs healthy cfDNA fragment size distribution published after 2016 years. Most of the pre-omics era studies on distinguishing cancer to healthy individuals by fragment size distribution were based on the q-PCR analysis, while this is not mentioned. The first observation was made by Mouliere et al. (PlosOne, 2011) who studied by fractional size distribution with nested Q-PCR systems "the size distribution profile of ctDNA fragments in plasma" and indicated that "the size distribution profile of ctDNA fragments can be used to discriminate between healthy and cancer patients". This team later showed that mutant fragments are shorter than wild type counterparts (Mouliere et al.; Transl.Oncol, 2013), Lo's group later confirmed this observation by showing strong but somewhat different differences in regards to size ranges Jiang et al. (PNAS, 2015) or more recently in Sun Kun et al. (Genome Res, 2019).

– Renaud et al. work as well than those cited in the manuscript such as Cristiano et al., relies on Mouliere et al. (PlosOne, 2011) first observation. Note, Cristiano et al. cited this report as well as Sanchez et al. (npgGenomic Medicine, 2018), Jiang et al., and Underhill (Plos Gen. 2015). Finally, Sanchez et al. (JCI insight, 2021) refined Mouliere et al. (2011) observation by using sWGS from double and single stranded DNA library preparations. The authors should necessarily add these citations.

– The authors showed exactly the same illustrative fragment size profile in Figure 1 (normal and tumor signature) as those presented earlier by Sanchez et al. (JCI insight, 2021). Consequently, they should at least mention in a few sentences the observations made by Jiang et al. and by Sanchez et al. (JCI insight, 2021) in order to validate their assumption.

3. In light of the previous concerns, the title should be corrected, as the term discovery is improper in regards to the study. The signatures were previously made; the method used here, solely allowed to combine fragment length markers in an optimal setting while developing a technology. This is already important since this method appears very performant, and the reviewer is impressed. Discovery should be replaced by another term such as detection or assessment.

*Reviewer #2 (Recommendations for the authors):*

Specific comments for authors to address:

1) In the introduction, it would be better to describe a bit more details about tumor DNA length characteristics that were reported previously such as https://journals.plos.org/plosgenetics/article?id=10.1371/journal.pgen.1006162, https://www.pnas.org/content/112/11/E1317, https://www.pnas.org/content/115/46/E10925, https://stm.sciencemag.org/content/10/466/eaat4921.abstract, therefore the readers would have a more comprehensive understanding of what is already known in the prior art and why the analysis of tumor-specific length profile in this study is important.

2) The authors should compare whether an NMF component would give a better correlation with tumor DNA estimated by ichorCNA, compared with the short DNA proportion (i.e.% of plasma DNA < 150 bp) in the overall size profile.

3) It seems a bit confused to the reviewer that the classification power between cases and controls improved as the number of components used in the NMF process. But the authors did not see the improvement in the correlation between tumor DNA fraction and the weight of one component reflecting the tumor contribution. Such observation should be discussed and clarified.

4) The AUC was calculated using repeated 10-fold cross-validation. It would be better to show the boxplot of AUC values for a given number of NMF components (Figure 3c).

5) It will be informative to estimate the minimum number of samples is required for performing an accurate NMF analysis.

*Reviewer #3 (Recommendations for the authors):*

1) The authors claim their model is directly measuring ctDNA load. Since an untargeted (unsu-pervised) model to assess the tumor fraction for monitoring purposes would be of great value, it would be interesting to see if changing levels of ctDNA can be captured by changing weights of the putative cancer Signature 1. Since their mCRPC cohort included patients with follow-up samples (at therapy start and progression) it is not clear why this has not been tested.

2) It should be mentioned that ichorCNA can only provide informative results at ctDNA frac-tion of 3% and higher. At higher tumor fractions the fragment signature signal might be rel-atively strong, explaining the good correlation, but what about samples with lower frac-tions? Please indicate the tumor fractions (median, min, max) assessed with ichorCNA and based on the VAFs of the identified mutations for all patients.

3) The authors claim that using NMF they were able to verify that the fragment length profile of the signature correlating with ctDNA levels does indeed match the length distribution of fragments containing mutations. However, although they included PBMCs in their mutation analysis the authors did not correct for clonal hematopoiesis. Recent studies showed that up to 20% of identified mutation are related to CH and that such mutations are also found in driver genes for solid tumors. Since there might be a size difference of mutated fragment originating from the tumor or PBMCs, respectively, this statement is overrated. In case their sequencing approach was not sensitive enough to detected CH-related variants this should at least be mentioned.

4) In Figure 2c, the shift towards smaller fragment is only seen for the dinucelosomes peak but not for the mononucleosomal fragments. How do the authors explain this (if not by contamination of CH-related variants)?

5) The authors demonstrate that NMF was more robust when less data is available than the ichorCNA algorithm, but they lacked to provide convincing data that NMF as a quantitative measure is more sensitive with respect to tumor fraction. To this end, in silico dilution ex-periments where they computationally mix healthy fragments and high ctDNA fraction sample(s) to a ctDNA fraction of their choice (with repetitions to achieve a more robust es-timation) should be performed.

6) Also, if there are no CNA or tumor fraction >3%, ichorCNA can literally not be used to de-termine ctDNA fraction, so the feasibility of the dilution approach to benchmark ichorCNA is not clear to me (in particular for early stage cases, in which most samples have tumor fractions >1%). Moreover, low depth can be overcome by additional sequencing whereas the tumor fraction cannot be enriched by more reads.

7) Using NMF the authors demonstrate that there are 2 major contributors to fragment length distribution and the addition of further signatures does not add anything. However, this statement is conflicting with results of the linear SVM classification case/control ap-proach (Figure 3c). A plateau of classification accuracy is reached when using around 30 signatures, which suggests this would be a good trade-off for number of signatures to reach high accuracy (AUC).

8) Figure 3d: Interestingly, open chromatin is associated with a higher fraction of dinucleosomal fragments. How do the authors explain this? Figure 3: I am not sure what the authors want to show here with Pearson's correlation coefficient, this does not look like a linear relationship.

9) The paragraph on "epigenetic fragment length" could be better presented. I would rephrase "epigenetic state". Both the DELFI ratio and the fragment length signature are surrogates for the chromatin state (open or closed). First the author talks about bin-wise ex-traction of two signatures. Why is it, that reducing bin size gives better classification results? ~50 kb bins seem to be best to predict ATACseq results. Is there literature that de-scribes open/closed chromatin in this way? Then they correlate the weights from that with ATACseq results. And finally, they use their bin-wise signature extraction results/weights as input for cancer/control classification. Could be split in two parts for clarity: one that talks about correlation with ATACseq results and one that discusses classification results

10) cfDNA fragment length profiles are particularly prone to pre-analytical (extraction, library prep) and analytical (clustering efficiency of various sequencings machines) factors. How do preanalyitcal/analytical factors affect the classification? The authors mention in the discussion that changes in the size profiles from their mCRPC cohort and the DELFI samples vary, which might be attributed due to such factors. Yet such confounding event will impact the robustness of such approaches and may prevent a widespread implementation of frag-mentomics. This should be further discussed.

11) How can the authors exclude that the above-mentioned differences are attributed to high-er/lower ctDNA fractions in the metastatic CRPC and the early stage DELFI samples? Even in Figure 2a it seems that samples with higher tumor fraction have more dinucelosomal frag-ments. Separate fragmentation profile plots for each stage should be shown; it would be interesting whether such profiles show any differences for stage I and stage II patients.

12) The authors could have checked in the DELFI cohort whether there is a (physiologic fragment length variation) between the seven cancer types? Would there always be the same two signatures coming up if the only consider one tumor type? Based on Supplementary Figure 4 this seems not to be case. In addition, there is a huge overlap of healthy individuals and cancer patients for signature 1? Please also present Supplementary Figure 4 with Signature 2. Can we really assume that this signature is the one related to cancer?

[Editors' note: further revisions were suggested prior to acceptance, as described below.]

Thank you for resubmitting your work entitled "Unsupervised discovery of fragment length signatures of circulating tumor DNA using Non-negative Matrix Factorization" for further consideration by *eLife*. Your revised article has been evaluated by Detlef Weigel (Senior Editor) and a Reviewing Editor.

The manuscript has been improved but there are some remaining issues that need to be addressed, as outlined below:

*Reviewer #1 (Recommendations for the authors):*

Essential revisions:

1. The authors partially answered to the concerns but did not add a flow chart to ease reading. In addition, they did not precise the pre-analytics such as delay between blood draw and plasma isolation, storage, centrifugation steps and speeds,… This would validate quantitative data accuracy and may influence fragment size.

2. Please replace "discovery" with "detection" in the title.

3. The authors should describe in more detail how they concluded that the NMF model can be reliably estimated (as well on which criteria) using as few as 20 samples.

4. The authors should indicate whether they have used a spiking strategy to evaluate whether their model is directly measuring ctDNA load, or whether they use another ctDNA techniques to compare. Otherwise, their evaluation is not complete, and this should be indicated in the limitations of the study in the discussion.

5. The authors answered to this concern by showing adequate data but omitting to discuss them. This is of importance since this cutoff appears as a critical limitation of the model.

6. Explanations from authors are reasonable. However, this discrepancy as well as these explanations should be inserted in the text.

7. The authors weakly addressed this concern. In particular, one of the major criticism of the Cristiano work is the selection of the healthy individuals: Are they age matched? Are they fully healthy? Are their plasma absolutely examined as those of cancer individuals? Any difference about pre-analytics or analytics between cancer types? Please discuss. Finally, please be cautious about fragment size cancer signature specificity.

---

## [Author Response]

Essential revisions:1. More information on the source of cancer patient plasma, characteristics of the patients, especially in respect to treatments at time of blood sampling, should be included. A flow chart summarising the overall results might be useful.

The timing of blood samples has been clarified in the results and methods sections of the manuscript. In brief, most samples were collected either before starting first line treatment or following relapse after first line treatment with a small minority of samples collected following relapse from later treatments.

2. The authors can consider citing some of the older work analyzing cfDNA fragment size distribution using PCR-based approaches.

We thank the referee for pointing out the lack of representation of the PCR-based studies of the cfDNA fragment length distribution in our introduction. We have now updated the introduction section to also cover four PCR-based studies.

3. The authors showed exactly the same illustrative fragment size profile in Figure 1 (normal and tumor signature) as those presented earlier by Sanchez et al. (JCI insight, 2021). Consequently, they should at least mention in a few sentences the observations made by Jiang et al. and by Sanchez et al. (JCI insight, 2021) in order to validate their assumption.

Again, we thank the referee for pointing out the missing references to two sequencing-based studies of cfDNA fragment length signals in cancer. We have updated the introduction and Results sections with references to both Jiang *et al.,* 2016 and Sanchez *et al.*, 2021.

4. As one could argue that the signatures of cfDNA fragment length had been discovered through various previously published work, in light of the previous concerns, term 'discovery' is perhaps inappropriate. Hence, we would suggest replacing the word 'discovery' by another term such as 'detection' or 'assessment'.

It was not our intention to indicate that we were the first to “discover” fragment length signatures of circulating tumor DNA. Instead, we wished to indicate that our method was able to discover tumor signature(s) without using e.g. mutations or xenografts for identifying tumor-derived cfDNA fragments. We have updated the title, introduction and Discussion sections to better reflect this.

5. In the introduction, it would be better to describe a bit more details about tumor DNA length characteristics that were reported previously, so that readers would have a more comprehensive understanding of what is already known in the prior art and why the analysis of tumor-specific length profile in this study is important.

We agree with the referee that prior art was insufficiently represented in the previous submission. We have included several additional references to both PCR and sequencing based studies of the cfDNA fragment length distribution in cancer. Finally, we have now included a more thorough motivation for our NMF model as an important contribution to the field. In brief, previous studies have relied either on xenografts or detailed knowledge about mutations e.g. from targeted sequencing to accurately determine the tumor fragment length distribution. NMF enables us to do this in a completely unsupervised manner and hence independently of mutational or clinical information.

6. The authors should compare whether an NMF component would give a better correlation with tumor DNA estimated by ichorCNA, compared with the short DNA proportion (i.e.% of plasma DNA < 150 bp) in the overall size profile.

We tested using the ratio of short-to-long fragments (100-150bp vs. 151-220bp) proposed by Cristiano et al. This yielded a correlation with the VAF based estimates of ctDNA fraction of 0.68 on the shallow WGS dataset and hence significantly lower than the NMF based estimate (r=0.75). We have added a comment to the main text and a supplementary figure (Figure 2 —figure supplement 3).

7. It seems somewhat intriguing that the classification power between cases and controls improved as the number of components used in the NMF process. However, the authors did not see an improvement in the correlation between tumor DNA fraction and the weight of one component reflecting the tumor contribution. These observations should be discussed and clarified in a revised manuscript.

On the mCRPC data, using more than two components does in fact improve the correlation – albeit only slightly (two components: 0.75, three components: 0.78, four components: 0.79; Figure 2 —figure supplement 2). Using more than two components involves a supervised element as we need to find out which components are tumor associated and sum up their contributions. In order to avoid this, and also to have a more interpretable model, we settled on the model with two components. On the DELFI data, two components were clearly insufficient. We attribute this to greater diversity in the DELFI data due to the many different cancer types included as well as potential technical differences. We have clarified this in both the results and Discussion sections of the manuscript.

8. The AUC was calculated using repeated 10-fold cross-validation. It would be better to show the boxplot of AUC values for a given number of NMF components (Figure 3c).

We now show boxplots to reveal the variation in AUC values between repeats.

9. It will be informative to estimate the minimum number of samples is required for performing an accurate NMF analysis.

We have now included tests of the robustness of our method to the number of samples. Across a range of dataset sizes, we repeatedly sampled a dataset with the appropriate number of samples and trained an NMF model. For each trained model, we then compared the estimated signatures with those estimated using the full datasets. Using this approach, we found that the NMF model can be reliably estimated using as little as 20 samples. These results have been incorporated in the Results section and a new supplementary figure (Figure 2 —figure supplement 5).

10. The authors claim their model is directly measuring ctDNA load. Since an untargeted (unsupervised) model to assess the tumor fraction for monitoring purposes would be of great value, it would be interesting to see if changing levels of ctDNA can be captured by changing weights of the putative cancer Signature 1. Since their mCRPC cohort included patients with follow-up samples (at therapy start and progression) it is not clear why this has not been tested.

We thank the referee for suggesting to use serially sampled data to assess our method. Unfortunately, all the samples from the mCRPC cohort were collected either at baseline (i.e. before treatment) or upon progression with no samples collected at response. We thus do not have a clear a priori expectation of the direction of change of the ctDNA fraction across time and hence cannot use serial information to evaluate our method. Serially collected samples are also available in the DELFI cohort. A small subset of the subjects in the DELFI cohort have serial samples that were also examined using targeted sequencing in a previous publication. We have now added a supplementary figure that compares the prediction probabilities obtained by our classification method using 30 fragment length signatures to the maximum driver mutation allele fraction observed in the targeted sequencing (Figure 3 —figure supplement 4).

11. It should be mentioned that ichorCNA can only provide informative results at ctDNA fraction of 3% and higher. At higher tumor fractions the fragment signature signal might be relatively strong, explaining the good correlation, but what about samples with lower fractions? Please indicate the tumor fractions (median, min, max) assessed with ichorCNA and based on the VAFs of the identified mutations for all patients.

We have added the requested information about ichorCNA and VAF-based ctDNA% estimates in the mCRPC dataset both as summary statistics (median, min, max) in the Results section and as violin plots in the supplement (Figure 2 —figure supplement 1).

12. The authors claim that using NMF they were able to verify that the fragment length profile of the signature correlating with ctDNA levels does indeed match the length distribution of fragments containing mutations. However, although they included PBMCs in their mutation analysis the authors did not correct for clonal hematopoiesis (CH). Recent studies showed that up to 20% of identified mutation are related to CH and that such mutations are also found in driver genes for solid tumors. Since there might be a size difference of mutated fragment originating from the tumor or PBMCs, respectively, this statement is overrated. In case their sequencing approach was not sensitive enough to detected CH-related variants this should at least be mentioned.

We do make sure that the ctDNA estimates from targeted sequencing are not biased by clonal hematopoiesis. During variant calling, we compare the cfDNA sequence data with sequence data from PBMCs. And after variant calling, we manually inspect all the driver mutations that are used to estimate ctDNA fraction and make sure that no alternative reads are seen in the PBMC data. The PBMC sequence data has a mean coverage of 306X (range: 253-728X). We have now added the mean PBMC sequence depth to the methods section and have specified that we check that no alternative reads are seen in the germline during the manual inspection of the driver variants.

13. In Figure 2c, the shift towards smaller fragment is only seen for the dinucelosomes peak but not for the mononucleosomal fragments. How do the authors explain this (if not by contamination of CH-related variants)?

The reviewers are correct that the shift towards lower fragment sizes is more pronounced for the nucleosomal peak but we do also see a small shift in the mononucleosome peak. This plot is only based on reads that overlap known driver mutations and is therefore based on a small number of fragments from a few positions in the genome, which explains the differences between the global fragment length distributions and the distributions seen in this plot. As explained in the response above, we are quite certain that the mutations we look at are not CH-related variants. A possible explanation for the differences in the fragment length distributions is that the global distribution could reflect that the ctDNA fragments tend to come from other parts of the genome than the non-cancer cfDNA fragments but in Figure 2c, we restrict the fragments to come from the same positions.

14. The authors demonstrated that NMF was more robust when less data was available than the ichorCNA algorithm, but they lacked to provide convincing data that NMF as a quantitative measure is more sensitive with respect to tumor fraction. To this end, in silico dilution experiments where they computationally mixed healthy fragments and high ctDNA fraction sample(s) to a ctDNA fraction of their choice (with repetitions to achieve a more robust es-timation) should be performed.

We agree with the referee that *in silico* dilution experiments would be informative regarding the robustness of NMF versus other methods at different ctDNA levels. Unfortunately, we do not have any healthy controls available for the mCRPC cohort and hence cannot perform the requested experiments. We have added a comment about this limitation to the discussion and have also made it clear that more experiments will be needed in order to determine which method, NMF or ichorCNA, is most sensitive.

15. If there are no CNA or tumor fraction >3%, ichorCNA can literally not be used to determine ctDNA fraction, so the feasibility of the dilution approach to benchmark ichorCNA is not clear to the reviewer (in particular for early-stage cases, in which most samples have tumor fractions >1%). Moreover, low depth can be overcome by additional sequencing whereas the tumor fraction cannot be enriched by more reads.

As the vast majority of samples have ctDNA levels above 3% as judged using driver VAFs and as our subsampling experiments do not change this, we believe our conclusions still stand. We agree with the referee that additional sequencing can overcome low depth but not low tumor fraction. A more ideal experiment would be to instead lower the ctDNA fractions by *in silico* dilution as also suggested by a referee in comment #14. Unfortunately, no healthy controls were available in the mCRPC cohort and hence we had instead to rely on subsampling experiments to prove the robustness of methods.

16. Using NMF the authors demonstrate that there are 2 major contributors to fragment length distribution and the addition of further signatures does not add anything. However, this statement is conflicting with results of the linear SVM classification case/control approach (Figure 3c). A plateau of classification accuracy is reached when using around 30 signatures, which suggests this would be a good trade-off for number of signatures to reach high accuracy (AUC).

This questions was also raised by another referee (comment #7) and we agree that our presentation of results regarding the optimal number of NMF components was not sufficiently clear in the previous version of the manuscript. This has now been clarified in the manuscript. In brief, using more than two components on the mCRPC data did not yield significant improvement in ctDNA correlation, whereas the gain was much bigger on the DELFI data. We speculate that this is due to higher heterogeneity (more cancer types, individuals) in the DELFI data.

17. Figure 3d: Interestingly, open chromatin is associated with a higher fraction of dinucleosomal fragments. How do the authors explain this? Figure 3: the reviewer is not sure what the authors want to show here with Pearson's correlation coefficient, this does not look like a linear relationship.

We have surveyed the literature to see if previous articles have mentioned that open chromatin is associated with a higher fraction of dinucleosomal fragments but have not found any previous reports of this. We agree that it is an interesting observation but unfortunately we do not have a good explanation for why it should be the case.

We agree that Figure 3e did not show a linear relationship and have now changed the figure and the text so that we use Spearman's rank correlation coefficient instead of Pearson’s correlation coefficient.

18. The paragraph on "epigenetic fragment length" could be better presented. I would re-phrase "epigenetic state". Both the DELFI ratio and the fragment length signature are surrogates for the chromatin state (open or closed). First the author talks about bin-wise extraction of two signatures. Why is it, that reducing bin size gives better classification results? ~50 kb bins seem to be best to predict ATACseq results. Is there literature that describes open/closed chromatin in this way? Then they correlate the weights from that with ATACseq results. And finally, they use their bin-wise signature extraction results/weights as input for cancer/control classification. Could be split in two parts for clarity: one that talks about correlation with ATACseq results and one that discusses classification results.

We appreciate the suggestions for improving the section and have tried to do so. Instead of writing about “epigenetic state” we now use “chromatin state” and we have split the section in two as suggested. With regards to why a smaller bin size leads to better classification results we can see that the ATACseq values do fluctuate to some degree from bin to bin even with the smaller bins sizes. So if the classification results are achieved through prediction of chromatin state then it makes sense that the smaller bins that better capture the small range changes in chromatin state would be best. We are not aware of any existing literature besides the 2019 DELFI article that deals with this issue.

19. cfDNA fragment length profiles are particularly prone to pre-analytical (extraction, library prep) and analytical (clustering efficiency of various sequencings machines) factors. How do preanalyitcal/analytical factors affect the classification? The authors mention in the discussion that changes in the size profiles from their mCRPC cohort and the DELFI samples vary, which might be attributed due to such factors. Yet such confounding event will impact the robustness of such approaches and may prevent a widespread implementation of fragmentomics. This should be further discussed.

We agree with the referee that technical factors may significantly impact the fragment length distribution and hence make it hard for fragment length based models to generalize to new data with different technical characteristics. We have now added an additional statement regarding this limitation to the discussion.

20. How can the authors exclude that the above-mentioned differences are attributed to higher/lower ctDNA fractions in the metastatic CRPC and the early stage DELFI samples? Even in Figure 2a it seems that samples with higher tumor fraction have more dinucleosomal fragments. Separate fragmentation profile plots for each stage should be shown; it would be interesting whether such profiles show any differences for stage I and stage II patients.

Our statements regarding potential technical differences between mCRPC and DELFI were based on comparing the estimated NMF signatures (e.g. Figure 2b vs Figure 3b), which – hopefully – eliminates the effect of different ctDNA levels. That said, it cannot be ruled out that the large differences in ctDNA levels between mCRPC and DELFI affects the signature estimates or that other, biological differences (e.g. stage effects) contribute. We have now clarified this in the manuscript and included a “spaghetti plot” showing fragment length distribution for the different stages in the DELFI data (Figure 3 —figure supplement 1). The stage plot confirms differences across stages, albeit subtle, which likely reflect increased ctDNA% at higher stages.

21. The authors could have checked in the DELFI cohort whether there is a (physiologic fragment length variation) between the seven cancers types? Would there always be the same two signatures coming up if the only consider one tumor type? Based on Supplementary Figure 4 this seems not to be case. In addition, there is a huge overlap of healthy individuals and cancer patients for signature 1? Please also present Supplementary Figure 4 with Signature 2. Can we really assume that this signature is the one related to cancer?

To address this, we reran NMF separately for each cancer type. The AUC was highest for colorectal cancer patients (0.93) and lowest for gastric cancer (0.56). Full results are shown in Figure 3 —figure supplement 3a. Furthermore, we sought to determine whether the inferred signatures differ from one cancer type to the next. The different inferred signatures were plotted (Figure 3 —figure supplement 3b). Apart from gastric cancer, which happens to be the one with the lowest AUC, most fragment length signatures do not seem to change depending on the cancer type.

22. Abbreviations should be defined at the beginning for better readability.

Done.

23. Resolution of figures need to be improved. Comprehensive legends should also be included for Supplementary figures.

We have now uploaded all figures as good quality pdf figures instead of just the low-quality inlined figures. Furthermore, we have improved the appearance of the supplementary figures and provided them with comprehensive legends.

24. Matching colors for better readability are recommended.

We have now tried to use the same color scheme in all figures.

25. Specify average read pairs for cfDNA samples and corresponding PBMCs.

We have now added a summary of the number of reads pairs for all samples sequencing in the mCRPC cohort.

26. Excel Supplementary file: Supplementary table 2 should be mentioned in the manuscript.

We thank the referee for pointing out the omission of these references. Supplementary tables 1 and 2 contain cfDNA fragment length counts for all samples for sWGS and targeted sequencing, respectively. Furthermore, VAF- and CNV-based estimates of the ctDNA% are included in both tables. References to the two tables have now been included in the manuscript.

27. ctDNA (%) on the x-axis in Figures 2 is calculated from targeted mutation analysis, right? How was est.ctDNA % in the Supplementary figures 2&3 calculated?

We apologize for not making this sufficiently clear. All ctDNA% estimates used for validation were based on VAFs from targeted mutation analyses unless it is explicitly specified that it is based on CNVs / ichorCNA. We have now clarified this in all figures.

28. What is the original h matrix in NMF in Figure 2f? There is a normal h matrix described as weight matrix, but I haven't seen the definition of what with "original" h matrix is meant.

“Original h matrix” referred to using signatures estimated on the full dataset to estimate the weights on the subsampled data. We agree with the referee that this was not clear in the previous version of the manuscript. We have now removed that line from Figure 2f in order to improve readability of the subsampling analysis.

29. Assumptions for linear correlation using Pearson are not mentioned, I don't know whether they are met at all. Looking at the figure it does not seem like they are met. Normality of the variables should at least be checked.

We thank the reviewers for bringing this omission to our attention. Since the results in Figure 3e do not show a linear relationship we have now changed the figure and the text so that we use Spearman's rank correlation coefficient instead of Pearson’s correlation coefficient.

30. How is NMF performing on previously unseen data? This part is missing. The authors could have checked for overfitting in this way for example calculating the reconstruction error on a test dataset.

To address this, we modified our subsampling analysis to repeatedly divide the data into two, a training and testing set. The average correlation of the weights estimated on the training set with ctDNA% was 0.75, whereas the average correlation on the testing set was 0.74 thus indicating that the NMF is robust to novel data (Figure 2 —figure supplement 6).

31. It is not clear whether cross validation was used only for the classification task or also to evaluate the NMF decomposition. Are those signatures representative?

We do not retrain the NMF components in each fold in the cross validation. The results mentioned in the response to question 9 and shown in Figure 2 —figure supplement 5 show that signatures inferred from 90% of the data are indistinguishable from signatures inferred from the whole data set.

32. It would be nice to see the position of the excluded bins based on +- 2* IQR. They already excluded the ENCODE blacklist regions. Other outliers may be real biological signal especially in some type of cancer where copy number alteration play an important role.

We exclude bins that are outliers with regards to depth in controls. It is the same bins that are removed in all samples and we do not look at the case samples when we determine which bins to remove. So biological signals from copy number alterations in the cancer samples do not affect which bins are removed.

[Editors' note: further revisions were suggested prior to acceptance, as described below.]

The manuscript has been improved but there are some remaining issues that need to be addressed, as outlined below:Reviewer #1 (Recommendations for the authors):Essential revisions:1. The authors partially answered to the concerns but did not add a flow chart to ease reading. In addition, they did not precise the pre-analytics such as delay between blood draw and plasma isolation, storage, centrifugation steps and speeds,… This would validate quantitative data accuracy and may influence fragment size.

We agree with the referee that the technical description of cfDNA processing was not completely clear in the previous version of the manuscript and we have now added extra details regarding this to the methods section. The description in the “Sample processing and DNA extraction” subsection now includes all the requested details about the pre-analytics. Besides the main overview in Figure 1 we have now added a flow chart giving an overview of the different analyses and experiments presented in Figure 2 and Figure 3 as supplementary figures to Figure 2 and Figure 3 respectively.

2. Please replace "discovery" with "detection" in the title.

The title has been changed as requested.

3. The authors should describe in more detail how they concluded that the NMF model can be reliably estimated (as well on which criteria) using as few as 20 samples.

We agree with the referee that the subsampling procedure and criteria used for assessing reliability was not clear enough in the previous version of the manuscript. We have now added more thorough descriptions to both the results and methods sections. The methods section now includes a subsection called “Subsampling experiments” containing the details about the robustness experiments presented in Figure 2f, Figure 2 supp. 5 and Figure 2 supp.6.

4. The authors should indicate whether they have used a spiking strategy to evaluate whether their model is directly measuring ctDNA load, or whether they use another ctDNA techniques to compare. Otherwise, their evaluation is not complete, and this should be indicated in the limitations of the study in the discussion.

We agree with the referee that using spike-ins could have been a valuable add on to the present study, however this was not done due to lack of healthy controls for the mCRPC cohort. We have clarified this in the Discussion section of the manuscript. Please see the response to questions #11 for more details.

5. The authors answered to this concern by showing adequate data but omitting to discuss them. This is of importance since this cutoff appears as a critical limitation of the model.

We have added the following text to the discussion acknowledging the limitations of the current study:

“However, the lack of low ctDNA samples is a limitation of the current study and further experiments either based on samples with lower tumor burdens, spike-ins or in silico dilution are needed to confirm this assertion as it could not be directly tested with the data available in this study. A lack of healthy controls for the mCRPC cohort data means that we were not able to perform spike-in or in silico dilution experiments in this article”

6. Explanations from authors are reasonable. However, this discrepancy as well as these explanations should be inserted in the text.

We believe that the text added in response to the previous question covers these concerns as well.

7. The authors weakly addressed this concern. In particular, one of the major criticism of the Cristiano work is the selection of the healthy individuals: Are they age matched? Are they fully healthy? Are their plasma absolutely examined as those of cancer individuals? Any difference about pre-analytics or analytics between cancer types? Please discuss. Finally, please be cautious about fragment size cancer signature specificity.

We agree with the referee that technical or biological factors may have affected our results. We have now added the statement below to the discussion:

“A number of technical and biological factors in the DELFI study design may have inflated our classification performance. For instance, cases were generally older than controls and hence the observed differences in the fragment length distribution between cases and controls may partially reflect underlying comorbidities associated with higher age.”

We are unaware of any specific differences in how samples from cases and controls were processed in the DELFI study and hence feel unable to comment further on this in the manuscript.